# Valproic Acid Prodrug Affects Selective Markers, Augments Doxorubicin Anticancer Activity and Attenuates Its Toxicity in a Murine Model of Aggressive Breast Cancer

**DOI:** 10.3390/ph14121244

**Published:** 2021-11-30

**Authors:** Nataly Tarasenko, Harel Josef Wilner, Abraham Nudelman, Gania Kessler-Icekson, Ada Rephaeli

**Affiliations:** 1Felsenstein Medical Research Center, Sackler Faculty of Medicine, Beilinson Campus, Tel Aviv University, Petach-Tikva 49100, Israel; nataliytarasenko@gmail.com (N.T.); harel.j.wilner@gmail.com (H.J.W.); icekson@post.tau.ac.il (G.K.-I.); 2Chemistry Department, Bar-Ilan University, Ramat-Gan 52900, Israel; nudelman@biu.ac.il

**Keywords:** triple-negative breast cancer, valproic acid prodrug, doxorubicin, DNA damage, fibrosis

## Abstract

We studied the unique inhibitor of the histone deacetylases (HDAC) valproate-valpromide of acyclovir (AN446) that upon metabolic degradation release the HDAC inhibitor (HDACI) valproic acid (VPA). Among the HDAC inhibitors that we have tested, only AN446, and to a lesser extent VPA, synergized with doxorubicin (Dox) anti-cancer activity. Romidepsin (Rom) was additive and the other HDACIs tested were antagonistic. These findings led us to test and compare the anticancer activities of AN446, VPA, and Rom with and without Dox in the 4T1 triple-negative breast cancer murine model. A dose of 4 mg/kg once a week of Dox had no significant effect on tumor growth. Rom was toxic, and when added to Dox the toxicity intensified. AN446, AN446 + Dox, and VPA + Dox suppressed tumor growth. AN446 and AN446 + Dox were the best inhibitory treatments for tumor fibrosis, which promotes tumor growth and metastasis. Dox increased fibrosis in the heart and kidneys, disrupting their function. AN446 most effectively suppressed Dox-induced fibrosis in these organs and protected their function. AN446 and AN446 + Dox treatments were the most effective inhibitors of metastasis to the lungs, as measured by the gap area. Genes that control and regulate tumor growth, DNA damage and repair, reactive oxygen production, and generation of inflammation were examined as potential therapeutic targets. AN446 affected their expression in a tissue-dependent manner, resulting in augmenting the anticancer effect of Dox while reducing its toxicity. The specific therapeutic targets that emerged from this study are discussed.

## 1. Introduction

Acetylation of histones plays an important role in cancer development and progression. A hallmark of aggressive tumors is the disruption of the balance between histone acetyltransferases and histone deacetylases (HDACs), Currently, four HDACs have been approved by the US FDA [1,2]. The disrupted expression of HDACs in cancer cells changes their epigenetics and protects them from genotoxic stress, granting them a survival advantage. HDAC isoenzymes class I have been shown to be strongly expressed in more aggressive breast cancer (BC) types [3,4]. A structurally diverse group of HDAC inhibitory molecules exhibiting pleiotropic cytotoxic effects on various cancer cells in vitro and in vivo were described [1,2,3,4]. Among them, SAHA, belinostat, panobinostat, and the cyclic peptide, romidepsin, were approved by the FDA for the treatment of cancer [1,2]. The HDAC inhibitor valproic acid (VPA), alone and in combination with anticancer agents, has exhibited therapeutic activity in various cancer indications including metastatic BC [5,6]. Our research has centered on prodrugs of aliphatic HDAC inhibitors for the treatment of cancer and, cardiovascular diseases and anemia [7,8,9,10,11,12,13,14]. The butyric acid prodrugs AN9 (pivaloyloxymethyl butyrate) and AN7 (butyroyloxymethyl diethyl phosphate) exhibited anticancer and cardioprotective activities in preclinical studies [10,13]. AN9 showed safety and improvement in the well-being of cancer patients in clinical trials Phases I and II [15,16]. AN7, a water-soluble butyric acid (BA) prodrug, was found to be a significantly better anticancer drug than AN9 [17]. Next, we synthesized and tested VPA derivatives as anticancer agents. AN452 the VPA ester of acyclovir, AN463 the VPA amide (valpromide) of acyclovir, and AN446 the amide-ester (valpromide-ester) of acyclovir. AN463, shown to be inactive and the ester AN452 was 5–10-fold less active than AN446 [10]. In vitro anticancer activities of AN446 were shown to be 2–5-fold more potent than AN7, and >100-fold more potent than VPA; AN446 crosses the blood–brain barrier and is orally bioavailable [10,11]. AN446 specifically affected cancer cells compared to normal cells and normal tissues, in vitro and in vivo [11,12,13]. AN446 reduced toxicity of doxorubicin (Dox) to astrocytes and cardiomyocytes and exhibited synergistic interaction with Dox in triple-negative breast cancer (TNBC) and in glioblastoma U251 cells [12,13].

As an HDAC inhibitor, AN446 induced a tissue-unique pattern of acetylation and methylation of histones and gene expression in tumors compared to heart tissues [13]. In addition, we showed in glioblastoma cells and xenografts bearing glioblastoma tumors that AN446 possessed anticancer efficacy and improved the efficacy of Dox, while at the same time protecting the mice from Dox-induced cardiotoxicity [11].

AN446 properties qualify it as a potential candidate for the treatment of TNBC. Advances in the treatment of BC have been attributed largely to early detection and identification of specific therapeutic targets, such as estrogen receptor (ER), human epidermal growth factor receptor 2 (HER2), and progesterone receptor (PR). This led to targeted therapy and increased patient survival. However, the TNBC subgroup lacks detectable levels of ER, PR, and HER2, leaving few treatment options, mainly chemotherapy. This high-risk subgroup of patients has the worst survival rate (~20%). Many TNBC patients experience treatment-related morbidity and early relapse [18]. Therefore, identifying specific therapeutic targets in this subgroup of patients is imperative. The first-line protocols for TNBC include Dox treatment, which is complicated by dose-limiting toxicity, particularly cardiotoxicity [18].

AN446, which synergizes with Dox and has exhibited anticancer and cardioprotective properties, may be a promising candidate to treat TNBC. We evaluated the potential of AN446 compared to leading HDAC inhibitors, as single agents and in combination with Dox, for the treatment of TNBC in vitro and in the TNBC mouse model. In addition to evaluating anticancer and anti-metastatic activities, we also examined the effect of the treatments on vital organs.

The disruptive expression of HDACs in TNBC is specifically targeted by HDAC inhibitors, offering additional potential targets in TNBC [4]. A promising candidate is the population of cancer-associated fibroblasts that is prominent in metastatic TNBC, where they support the tumor and metastasis. Therefore, targeting tumor fibrosis could lead to disruption of the tumor supportive microenvironment, and, consequently, metastasis [19]. Other targets examined were representative genes that control and regulate tumor growth, DNA damage, ROS production, and inflammation. Along with the evaluation of anticancer and anti-metastatic activities, we examined the treatment effects on vital organs as well.

## 2. Results

### 2.1. Characterization of the Interaction between Dox and HDACIs in Murine TNBC Cells

The advantages of combination therapy in the treatment of advanced cancer are well recognized [20]. Since Dox is included in most treatment protocols of TNBC, and HDAC class I enzymes are potential specific targets, we searched for effective combinations of leading HDACIs that would augment the therapeutic impact of Dox. Prior to in vivo studies, we examined the nature of the interaction between the leading HDACIs to Dox on murine TNBC 4T1 cells. This was evaluated by MEA (Table 1). As shown previously, AN446 was highly synergistic with Dox (CI of 0.6) [11]. Herein, we have shown that VPA was slightly synergistic (CI of 0.9), and ROM was additive (CI of 1), panobinostat (CI of 1.1), SAHA (CI of 1.3), entinostat (CI of 1.6), and belinostat (CI of 1.6) were antagonistic (Figure 1).

### 2.2. In Vivo Efficacy of HDACIs as Single Treatment and in Combination with Dox in Mice Model of TNBC

Based on their properties in the combination study, the highly synergistic combination of AN446 + Dox, the weakly synergistic combination of VPA + Dox, and the additive Rom + Dox were selected for in vivo comparative studies. Based on our previous studies [9,10,11,12], literature reports, and preliminary experiments to examine the efficacy and toxicity of the tested drugs, the efficacious and maximum tolerated drugs doses found were chosen for this experiment. These drugs, alone or in combination with Dox (Figure 2), were administered to Balb-c mice bearing sc (subcutaneously) 4T1 tumors. Rom treatment was initiated at 2 mg/kg per week, and after a week, due to toxicity, spatially in the group treated with Rom + Dox, was reduced to 1 mg/kg per week. The toxicity was evidenced by body-weight reduction, appearance, and mortality.

When the experiment was terminated after 26 treatment days, only 50% of the mice in the vehicle-treated group survived. In the group that received AN446, 25 mg/kg thrice weekly, 90% of the mice survived. In the group treated once a week with 4 mg/kg Dox, only 40% survived. The addition of AN446 to Dox increased their survival from 40% to 80% (Figure 2A). Mice treated with Dox or Rom as single agents exhibited reduced vitality, but their survival was not reduced significantly—possibly because of the short follow-up time. The only group whose survival was reduced significantly was treated with Dox + Rom: only 20% of the mice survived to the end of the experiment. The reduction in survival in this group was significant compared to all the tested groups (*p* < 0.01 by log rank analysis with Bonferroni’s correction for multiple comparisons). Therefore, Dox + Rom treatment was the most toxic.

Tumor growth measurements showed no significant differences between vehicle-treated mice and mice treated with VPA or Dox as single agents (Figure 2B). Significant delay in tumor growth was observed in mice treated with AN446 alone or Rom alone, or their combination with Dox, compared to the vehicle-, Dox-, or VPA-treated mice. The high toxicity of the combination of Rom + Dox (80% of the mice did not survive the treatment) can account in part for the delay in tumor growth in this group. The combination of VPA + Dox significantly delayed tumor growth and had a low toxicity. At the end of the experiment, the primary tumors were excised and weighed. Dox treatment had no significant effect on average tumor weight, while all other treatments significantly reduced tumor weight compared to the vehicle-treated control mice. Compared to Dox treatment, a significant decrease in average tumor weight was observed with AN446 (9 tumors), AN446 + Dox (8 tumors), VPA + Dox (8 tumors), and Rom + Dox (2 tumors) (Figure 2C). The results of tumor growth showed that AN446, AN446 + Dox, VPA + Dox, and Rom were effective in inhibiting tumor progression; as noted, the effect of Dox + Rom can be ascribed to its toxicity.

### 2.3. Effect of the Treatments on Tissue Fibrosis

The heterogeneous tumor tissue is comprised largely of cancer cells surrounded by fibroblasts, extracellular matrix, and blood vessels. This environment is essential to support tumor growth [21]. Therefore, inhibition of tumor fibrosis can be a viable therapeutic target to inhibit tumor growth. To that end, we examined the effect of the treatments on tumor fibrosis. Sections of fixed tumor tissues were stained with Picrosirius red and fast green to label fibrous collagen secreted by activated fibroblasts (Figure 3A). The following results were observed: high fibrosis in the tumors of vehicle-treated mice; the highest fibrosis in Dox-treated mice followed by Rom + Dox; significantly reduced fibrosis in tumors of mice treated with Rom, VPA, VPA + Dox, AN446, and AN446 + Dox compared to Dox-treated mice. While the addition of VPA or AN446 significantly reduced fibrosis in the tumor, the addition of Rom did not affect it. Fibrosis in tumors of mice treated with AN446 and AN446 + Dox was significantly lower than in vehicle and all other treatment groups, showing that AN446 was the most effective inhibitor of fibrosis.

Heart fibrosis is a hallmark of cardiomyopathy. Dox has been shown to induce heart fibrosis and cardiotoxicity [22,23]. The fixed heart tissue sections from the in vivo study were evaluated for fibrosis. The hearts of vehicle-treated mice were essentially fibrosis-negative displaying the lowest intensity of Picrosirius red and fast green staining. All the other treatments increased heart fibrosis, with Dox and Dox + Rom increasing it most significantly. Adding VPA or AN446 to Dox significantly attenuated Dox-induced fibrosis, whereas the addition of Rom to Dox treatment did not (Figure 3B). The influence of the drugs on kidney fibrosis resembled that of the heart. Kidneys of vehicle-treated mice were also mostly negative to Picrosirius red and fast green staining, while Dox and Dox + Rom treatments caused a significantly greater increase in fibrosis than all other treatments. (Figure 3C). The increased fibrosis in the heart and kidneys by Dox + Rom may explain this treatment toxicity. The addition of AN446 or VPA (to a lesser extent) to Dox treatment significantly attenuated Dox-induced fibrosis, which could in part explain the protective effects of VPA and AN446. The addition of Rom to Dox treatment did not reduce Dox-induced fibrosis (Figure 3C). As seen in the tumor and heart, AN446 was most effective in inhibiting Dox-induced fibrosis in the kidney.

### 2.4. Effect of Drug Treatment on the Lungs

Since metastases are expected to limit the air volume in the lungs, the area of the air gaps in sections of the lungs stained with H&E served as a measure of metastatic burden. The proliferation of cells in the lungs was assessed by immune-staining with Ki-67, a nuclear protein that is associated with cellular proliferation (Figure 4A). The air gaps were calculated as described in the Material and Methods and are shown in Figure 4A,B. High cell density, which limited air gaps, was visible in lung sections of the vehicle, Rom, and Rom + Dox-treated mice (Figure 4A,B). All the treatments except for those of Rom and Rom + Dox significantly increased air gaps. The increases in AN446-, AN446 + Dox-, and VPA + Dox-treated mice were significantly greater than in the lungs of vehicle- and Rom-treated mice. The air gaps in the lungs of AN446- and AN446 + Dox-treated mice were significantly greater than in Dox-treated mice. Collectively, all treatments except Rom inhibited the infiltration of cells to the lungs, but the greatest inhibition was exhibited by AN446 and AN446 + Dox. Consistent with these results, the morphology of the lungs of mice treated with AN446 and AN446 + Dox best resembled normal lung morphology. The epithelial cells surrounding the respiratory portion of the bronchial tree stained intensely with Ki-67, reflecting their renewal process. Disseminated Ki-67 staining was also visible in metastatic patches and most pronounced in the sections from vehicle-, Rom- and Dox-treated mice, indicating the presence of proliferating cells that can be cancer cells or fibroblasts. To distinguish between infiltrating cancer cells or fibroblasts, sections of the lungs were stained for fibrosis (Figure 4C). Collagen staining was visible as a thin layer of basement membrane associated with an epithelial layer of bronchi, blood vessels, and the alveolar interstitium. Pronounced collagen staining was visible in metastatic lung lesions from mice treated with Dox and Rom + Dox. The combination treatments of VPA + Dox and AN446 + Dox substantially reduced the fibrosis seen in Dox-treated mice. These findings are in agreement with the inhibitory effect of VPA and AN446 on the level of fibrosis in the tumors, hearts, and kidneys of Dox-treated mice.

In hearts and tumors of the treated mice, we compared the expression levels of key proteins known to be involved in cell proliferation, regeneration, antioxidative and anti-inflammatory activities, and DNA damage and repair, using Western blot analysis.

### 2.5. Effects of the Treatments on of c-MYC and SIRT 1 Expression

c-Myc and SIRT1 are commonly found in aggressive breast cancer tissues [24,25], making them potential therapeutic targets. The expression of c-Myc and SIRT1 in the tumors was significantly downregulated by AN446 and by the combination of AN446 and Dox. VPA alone significantly reduced the expression of SIRT1 and had no effect on c-Myc levels. The combination of VPA + Dox attenuated SIRT expression but was not statistically significant (Figure 5A, upper panels).

The changes induced in c-Myc and SIRT1 expression in the hearts differed from those seen in the tumors. The combination of AN446 + Dox significantly increased c-Myc expression [11], whereas Rom alone or in combination with Dox decreased it significantly. SIRT1 was upregulated by AN446 and AN446 + Dox, while all other treatments repressed its expression (Figure 5A, lower panels).

### 2.6. Effect of the Treatments on SOD1 and CoX2 in Tumors and Hearts of the Treated Mice

The expression in tumors of SOD1, an inducible enzyme implicated in cardioprotection [26], was not changed substantially by the various treatments. However, in the hearts of mice treated with AN446, as a single agent or in the combination with Dox, the expression of SOD1 was significantly elevated compared to all other treatments. This observation suggests that these two treatments imparted cardioprotection, partly due to increased SOD1expression.

The expression of the enzyme cyclooxygenase2 (COX2), also known as prostaglandin-endoperoxide synthase (PTGS), is associated with inflammation [27]. Only AN446 and AN446 + Dox significantly reduced COX2 expression in the tumors, while Dox alone induced a small, not significant increase in its expression. In the hearts, Dox and Rom + Dox significantly increased the expression of COX2, implicating them in the induction of inflammation and damage. As single agents, VPA and AN446 did not affect COX2 expression, but when VPA or AN446 was added to Dox, there was a significant downregulation of COX2 expression, suggesting that both VPA and AN446 inhibited Dox-induced fibrosis. (Figure 5B).

### 2.7. Effect of the Treatments on DNA Damage and Repair Response

One of the earliest events in response to DNA double-strand breaks (DSBs) involves the phosphorylation of histone H2AX protein (pH2AX) on serine 139, which is widely used as a tool to measure induced DNA DSBs [28]. The phosphorylation of H2AX on serine 249, a measure of induced double-strand breaks, significantly increased in the tumors by all treatments. The greatest increase was observed with AN446 and AN446 + Dox treatments: 10.4-fold and 13-fold, respectively. Rom, Rom + Dox, VPA, and VPA + Dox treatments induced 5.3, 8.6, 6.5, and 8.2-fold increase in pH2AX, respectively. In the hearts, Dox and Rom increased the expression of pH2AX, while VPA and AN446 as single agents did not affect it. The addition of AN446 or VPA to Dox attenuated pH2AX to a level significantly lower than that observed with Dox alone. AN446 was more effective in reducing pH2AX than VPA and brought it to a level that did not differ from that seen in vehicle-treated mice (Figure 6).

In response to DSB formation, pH2AX accumulates at the site and interacts with the repair complex that includes Rad51 protein, which plays a central role in homologous recombination-guided DNA repair of these lesions [29]. In the tumors, Dox induced the highest expression of Rad51, while all the other treatments—Rom, VPA, and AN446 as a single agent or in combination with Dox—suppressed Rad51 expression to a level significantly lower than that of vehicle-treated mice. These observations suggest that except for Dox as a single agent, all other treatments in the tumors added insult to injury by suppressing RAD51, which is essential for DNA repair.

In the hearts, the low levels of Rad51 expression were reduced by Dox alone and by the addition of Dox to all the other treatments, except for AN446. The addition of AN446 to Dox treatment led to a significant increase in Rad51 expression, suggesting that the cardioprotective effect of AN446 against Dox injury encompassed also increased DNA repair (Figure 6).

## 3. Discussion

The importance of finding a synergistic combination between HDAC inhibitors and Dox in TNBC rests upon the need to increase anticancer activity and reduce treatment toxicity. The rationale for drug combination treatment in cancer, and even more so in advanced metastatic TNBC, was demonstrated mathematically to offer a higher probability of cure as compared to monotherapy [20]. Dox is part of most TNBC treatment protocols [18]. We previously characterized the nature of the in vitro interaction between Dox and leading HDACIs in glioblastoma U251 and TNBC MDA-BA-231 human cell lines. In the former cells, only AN446 interacted synergistically, and in the latter cells, AN446 and VPA interacted synergistically with Dox, whereas entinostat, Rom, and SAHA interacted antagonistically [13].

To guide us in the selection of HDAC inhibitors for evaluation of anticancer activity and interaction with Dox, we employed the well-established 4T1 model developed in an immunocompetent (BALB/c) background as a preclinically relevant, spontaneously metastatic TNBC model. Initially, the interaction of leading HDAC inhibitors with Dox in this cell line showed that only VPA (weakly) and AN446 (highly) were synergistic with Dox, Rom was additive, and the other HDAC inhibitors were antagonistic. The difference between VPA and its prodrug AN446 in the interaction with DOX may be attributed to the efficient delivery of VPA by the prodrug that results in greater potency and reduced off-target toxicity. These observations underscore the potential benefit of treatment of TNBC with the combination of Dox and AN446. In addition, as we demonstrated earlier, the ability of AN446 to protect cardiomyoblasts and astrocytes from Dox toxicity grants it a unique advantage [11,12]. The inhibition of cancer cell growth and lower toxicity in noncancerous cells can be the basis for the superiority of AN446 over other HDAC inhibitors.

The above findings led to the in vivo testing of AN446, VPA, and Rom as single agents and in combination with Dox. While all combinations with Dox repressed tumor growth, the high toxicity and significant increase in mortality of Rom + Dox compared to all the other treatments, including the vehicle, rendered that combination intolerable. In contrast, the addition of AN446 or VPA to Dox lowered Dox toxicity, as evidenced by the increased survival of the animals. Tumor progression data was consistent with tumor weight gain at the end of the experiment. Except for the vehicle- and Dox-treated mice, all other treatments significantly inhibited tumor progression and tumor weight. The lowest weights of tumors were found in animals treated with VPA + Dox, AN446, and AN446 + Dox-, demonstrating that these combinations were the most efficacious. The tumors in the Rom + Dox-treated animals were significantly smaller, however, we cannot compare these results to those of other treatments as the treatment was highly toxic and only two mice survived it. Therefore, treatment with the combination of Dox and Rom was deemed intolerable.

Fibrosis in tumors provides a supportive environment for cancer cells. Growth factors secreted by cancer-associated cells stimulate tumor cell proliferation and increase blood supply to the tumor, enabling epithelial–mesenchymal transition (ETM) [30,31]. In contrast to our earlier findings that Dox reduced fibrosis in tumors of glioblastoma xenografts [12], in the 4T1 tumors in the present study, Dox both as a single agent and in combination with Rom or VPA significantly increased it. This finding is in line with Dox’s lack of anticancer activity, suggesting that the choice of treating TNBC with Dox should be re-evaluated. As single agents, Rom also increased tumor fibrosis, VPA had no effect and AN446 inhibited it. Fibrosis in mice treated with AN446 or with AN446 + Dox was dramatically reduced compared to vehicle-treated tumors. The reduction in the development of fibrosis and the consequent retardation of tumor growth in mice treated with VPA + Dox, AN446, and AN446 + Dox is in line with their anticancer activity [31].

Heart fibrosis is characterized by the expansion of the extracellular matrix and in particular the accumulation of collagen type I, a major player in fibrosis. Dox-induced damage in the heart leads to fibrosis, which reduces heart contractility and thereby its function [32]. Taken together with the low anti-cancer activity of Dox, this observation, lends further support to our suggestion to re-evaluate the use of Dox as a single agent for the treatment of TNBC. The heart fibrosis induced by Dox in our study was attenuated to a large extent by the addition of AN446 and to a lesser extent by VPA, demonstrating the ability of these agents to preserve the elasticity and functions of the heart.

Changes in the heart affect the hemodynamics of the kidneys, hence it is not surprising that inducing heart fibrosis led to similar but less dramatic changes in the kidney. The highest kidney fibrosis was induced by Dox, and, as in the heart, it was attenuated to a great extent by the addition of AN446 and to a lesser extent by the addition of VPA. Rom induced fibrosis similar to in the heart, and VPA and AN446 did not substantially induce it.

The main cause of high mortality rates due to breast cancer is the development of distant metastasis, which develops in the lungs in 60% of TNBC cases. When this occurs, life expectancy is low with a median survival of only 22 months [33]. The 4T1 syngeneic model enabled us to evaluate lung metastasis by measuring the decrease in air spaces in the lungs. Reduction in air space can be attributed to the presence of metastases or fibrotic tissue. The finding that the fibrotic area in the lungs co-resided in the cell-rich regions further supports the notion that air space reduction is the outcome of the accumulation of fibroblasts and of metastatic lung lesions. Therefore, reduction in air space in the lungs can be ascribed mostly to tumor metastases. While Dox-induced fibrosis in the heart and kidney reduced cellularity in the lungs and increased air gaps, strongly suggesting that though Dox did not significantly inhibit the primary tumor growth, it significantly inhibited lung metastasis. The benefits of AN446 rests in its ability to inhibit both primary tumor growth and the development of lung metastasis. VPA had similar effects as AN446 but to a lesser extent.

The underlying biochemical and molecular events associated with the changes in tumors and normal tissues provide insights into those changes and can point to new therapeutic targets. Myc is a master gene widely known to be elevated in the majority of aggressive cancers. It also controls the transcription of somatic cell proliferation and tissue regeneration. [11,34]. The tumorigenesis driven by elevated levels of Myc makes it a promising candidate to be an oncogenic-specific therapeutic target. However, because depressing its expression in all tissues can cause toxicity, treatment should be tissue-specific and affect mainly cancerous tissue. We showed that AN446 selectively and significantly reduced the elevated Myc level in the 4T1 tumors but not in the heart. Moreover, AN446 + Dox elevated Myc levels in the heart, suggesting that AN446 reverses the Dox-induced damage by elevating Myc expression, enabling regeneration of the injured tissue [11,34]. Previously we have shown that HDAC inhibitor elevated angiogenesis in the heart by downregulating the expression of the c-Myc-regulated angiogenesis inhibitor TSP-1 and co-elevated the expression of the proangiogenic factors FGF-2 and VEGF [9]. Similarly, we have shown that the addition of AN446 to Dox treatment, elevated angiogenesis in the heart by effectively and by a tissue specific-manner increased FGF-2 and VEGF expression [11], further supporting its cardioprotective function shown in this study. Similar to Myc, expression of SIRT1 was found to correlate with aggressive tumors [24], and to promote cancer progression in BC by modulating Akt activity [35], making it another potential therapeutic target. In our study AN446, AN446 + Dox, and to a lesser extent VPA, downregulated SIRT1 levels in the tumors. Finding that AN446 significantly reduces SIRT1 expression lends further support to its therapeutic potential for TNBC. Unlike in the tumors, AN446 and AN446 + Dox upregulated SIRT1 expression in the heart while the other treatments repressed it. Elevation of SIRT1 in the heart was previously shown to impart cardioprotection against oxidative stress, inflammation, and cardiac hypertrophy [36]. Dox is known to induce oxidative stress and inflammation leading to pathological conditions associated with cardiovascular diseases. The ability of Dox to dramatically suppress the expression of SIRT1 likely contributes to its cardiotoxicity, whereas SIRT1 upregulation by the addition of AN446 potentially helps attenuate Dox cardiotoxicity.

SOD1 is an anti-oxidative enzyme implicated in cardioprotection against Dox-induced toxicity [26]. While none of the treatments significantly affected its expression in the tumors, Dox reduced its expression in the heart and AN446, and AN446 + Dox elevated it. By elevating SOD1, a major scavenger of reactive oxygen species (ROS) in the heart, AN446 grants the heart a protective tool against oxidative stress-induced damage.

Oxidative stress can lead to inflammation, a process associated with COX2 activity. In the 4T1 tumors, Dox elevated the already high expression of COX2, further aggravating inflammation. AN446 significantly reduced inflammation and, when added to Dox, diminished the levels of inflammation associated with COX2 expression to the lowest level compared to all other treatments. Since COX2 is overexpressed in 40–50% of breast cancer patients [27], its repression by AN446 bears therapeutic importance for the treatment of BC. Elevated COX2 expression leads to tumor progression and metastasis due to multiple cellular events evoked by the increase in prostaglandins production in the tumor. The prostaglandin cascade plays a significant role in mammary carcinogenesis, such as the inactivation of host anti-tumor immune cells that increase the immuno-suppressor function of tumor-associated macrophages and the promotion of tumor cell migration and metastasis [37]. AN446 represses COX2, and therefore tumor progression. These observations support the notion that the selective targeting of COX2 by AN446 makes this protein a viable therapeutic target in aggressive BC.

In the heart, Dox and Rom increased COX2 expression, which can lead to acute and chronic inflammation [38,39]. VPA and AN446 as single-agent treatments had no significant effect, while in combination with Dox each of them imparted cardioprotection by significantly attenuating the Dox stimulation of COX2. Moreover, AN446 did this by significantly downregulating COX2 and significantly upregulating SOD1, leading to anti-inflammatory and anti-oxidative activities that culminated in robust cardioprotection.

Normal cells have developed mechanisms to cope with DNA damage, called the DNA damage response (DDR), that signal its presence and promote repair. Most cancers harbor defective DDR mechanisms that result in genome instability and enhanced tumor progression [40]. Using pH2AX as a marker for DNA damage and RAD51 expression as an indicator of DNA repair, we followed the effects of the drug treatments on these two opposing activities [41]. In the tumor, all treatments significantly increased DNA damage, with AN446 exerting the greatest effect. In the heart, AN446 exhibited the opposite effect: it inhibited DNA damage, and by elevating RAD51 expression it imparted cardioprotection against Dox-induced DNA damage. VPA also imparted protection against DNA damage but to a significantly lesser extent than AN446.

## 4. Material and Methods

### 4.1. Compounds and Reagents

AN446 was synthesized as described [10]; HDAC inhibitors—SAHA and entinostat (MS-275)—were obtained from Cayman Chemical (Ann Arbor, MI, USA), romidepsin (depsipeptide) from ApexBio Tech LLC (Houston, TX, USA), and panobinostat (LBH589) and belinostat (PXD101) from MedChem Express (Princeton, NJ, USA). These HDACIs were dissolved in DMSO to a concentration of 100 mM and stored in aliquots at −20 °C. Dox hydrochloride 2 mg/mL, was obtained from Ebewe Pharma Ges.m.b.H. (Unterach, Austria) and was diluted in saline solution. Polyclonal antibodies: c-Myc #9402, SIRT1 #2310 (Cell Signaling, Danvers, MA, USA) and superoxide dismutase 1(SOD1)#ab13498 (Abcam, Cambridge, UK), phospho-H2AX, Serine 139, pH2AX # A300-081A-M (Bethyl Laboratories, Montgomery, TX, USA), cyclooxygenase-2 (COX-2) #PA5-16817 (ThermoFisher, Rockford, IL, USA), Ki-67, BRB040 (Zytomed, Berlin, Germany). Mouse monoclonal antibodies: Rad51 #05-530-I (Sigma, Saint Louis, MO, USA), and actin # SKU:0869100-CF (MP Biomedicals, Aurora, OH, USA). Secondary antibodies: IRDye^®^ 680 goat anti-mouse or anti-rabbit IgG (LI-COR Biosciences, Lincoln, NE, USA), and biotinylated goat anti-rabbit IgG-B (Santa Cruz Biotechnology, CA, USA).

### 4.2. Cell Cultures

The murine mammary carcinoma 4T1 cells (ATCC^®^ CRL-2539™) were obtained from ATCC (Rockville, MD, USA). The cells were grown in RPMI-1640 with 100 units/mL penicillin, 100 µg/mL streptomycin, and 12.5 units/mL nystatin (Biological Industries, Beit Haemek, Israel), and incubated in a humidified atmosphere of 5% CO_2_ and 95% air at 37 °C. Cell viability assessment by the Hoechst assay was performed as described [42].

### 4.3. Murine 4T1 Breast Carcinoma Metastatic Model

Eight-week-old female BALB/c mice were purchased from Envigo (Envigo Rms, Inc., Jerusalem, IL, USA). 4T1 breast cancer cells (5 × 10^4^) were injected subcutaneously into their right flank. Treatments were initiated when the tumors reached ~40 mm^3^ (day 10 post-implantation). The mice were randomly divided into equal treatment groups (*n* = 10). Vehicle control (ip saline, thrice a week); intraperitoneal (ip) Dox 4 mg/kg, once a week (Dox); orally (po) AN446 25 mg/kg, thrice a week (AN446); ip Dox 4 mg/kg, once a week and po AN446 25 mg/kg, thrice a week (AN446 + Dox); ip romidepsin (Rom) 2 mg/kg, once a week, or as indicated; and ip Dox 4 mg/kg once a week, (Rom + Dox); po VPA 200 mg/kg, thrice a week (VPA); po VPA 200 mg/kg, thrice a week and ip Dox 4 mg/kg, once a week, (VPA + Dox). Tumor volumes were calculated according to the formula: (L × W2)/2, where the long axis (L) and short axis (w) were measured with an electronic digital caliper. The experiment was terminated after 26 days. The tumors, lungs, kidneys, liver, and hearts were harvested and weighed. Organs from three mice of each group were preserved for immunohistochemistry (IHC) and the remaining organs were frozen (−80 °C) for Western blot analyses.

### 4.4. Immunohistochemistry

Harvested organs were fixed in 4% paraformaldehyde for 24 h, washed with PBS, dehydrated in increasing alcohol concentrations, embedded in paraffin blocks, and processed as described [43]. Inactivation of avidin-biotin nonspecific binding was prevented by a blocking kit, according to the manufacturer’s protocol (Vector Laboratories, Burlingame, CA, USA). The sections were further incubated at 4 °C overnight with the Ki-67 antibody. The secondary antibody was biotin-conjugated goat anti-rabbit IgG (Santa Cruz, CA, USA). Slides were then stained with the ABC peroxidase system, developed with diaminobenzidine (DAB) (Vector Laboratories, Burlingame, CA, USA), and counterstained with hematoxylin (Bio-Optica, Milano, Italy). Slides were examined with an Olympus DP50 digital camera system [11].

### 4.5. Picrosirius Red and Fast Green Staining

Slides of selected tissues were stained with 0.1% Picrosirius red and 0.2% fast green FCF (Sigma), as described [22]. Collagen fibers appear red and non-collagen proteins green.

### 4.6. Western Blot Analysis

Protein levels in the samples were determined with the BCA protein assay kit (Pierce, Rockford, IL, USA), and the samples were subjected to Western blot analyses. Tumors and hearts were homogenized (Polytron; Kinematica, Lucerne, Switzerland) in a cold cell lysis buffer (containing 25 mM Tris-HCl, pH 7.5, 150 mM NaCl, 0.1% SDS, 0.1% sodium deoxycholate, 1.0% Triton X-100, and a protease inhibitor cocktail from Calbiochem, San Diego, CA, USA. After 30 min of incubation in ice, the samples were centrifuged at 12,500× *g* at 4 °C, and the supernatants were aliquoted and stored at −80 °C. The samples (30–45 μg protein per lane) were subjected to Western blot analyses. Separation of γH2AX (17 kDa), c-Myc (60 kDa), Rad 51 (37 kDa), SIRT1 (90 kDa), SOD 1 (65 kDa), and COX 2 (70 kDa) was performed on 15%, 12%, and 10% polyacrylamide gels, as indicated. The expression of proteins was visualized using their specified primary antibodies followed by the secondary IgG IRDye 680DX antibody [10]. Each detected band was quantified using the Odyssey Infrared Imaging System (LI-COR Biosciences, Lincoln, NE, USA) and normalized to the level of actin. The fold increase in a specific protein was determined by the ratio of the band intensity obtained from treated and untreated samples.

### 4.7. Data Analysis

Median effect analysis (MEA) was used to determine drugs’ interaction and combination index (CI). Drug concentration dependence plots were generated for each of the drugs as a single agent and in combination using CompuSyn software (CompuSyn, Inc. Paramus, NJ, USA). The IC_50_ of each tested compound was pre-determined (for the viability of the cells) in at least three independent experiments. The IC_50_ ratio of Dox in combination with its IC_50_ of Dox as a single agent determines the combination index (CI) in reducing the viability of cells [44]. A CI of 1, indicated additive effect, a CI < 1 indicated synergistic drugs interaction, and CI > 1 indicated antagonistic drugs interaction.

Mice survival was evaluated by log rank analysis with Bonferroni’s correction for multiple comparisons. Treatment failure (the experimental endpoints) was a loss of ≥20% body weight or a tumor volume of >1 cm^3^. The experiment was terminated on day 26 from the initiation of drug treatment. Tumor growth was analyzed by two-way ANOVA (Prism, GraphPad Software, San Diego, CA, USA).

### 4.8. Image Analysis

Pictures were captured at the specified magnification. Image analysis was carried out with the ImageJ software. All pictures went through color deconvolution using the color deconvolution plugin from Gabriel Landini (http://www.mecourse.com/landinig/software/cdeconv/cdeconv.html, accessed on 6 April 2020). The fibrosis area was measured by MRI Fibrosis Tool (http://dev.mri.cnrs.fr/projects/imagej-macros/wiki/Fibrosis_Tool, accessed on 1 November 2021). To identify air gaps, the same macro was applied to the lung images with a modified color vector. That is: the coloring was subtracted from the entire area of H&E-colored slides and filtered with the Color Adjust tool. Outliners were identified using the ROUT method with Q = 0.5. The D’Agostino and Pearson omnibus normality test was performed on each data set; for normally distributed data, one-way ANOVA was performed with Holm–Sidak’s multiple comparisons correction. When one of the treatment distributions was not normal, the non-parameter Kruskal–Wallis test was applied with Dunn’s test for multiple compression correction.

## 5. Conclusions

We have shown here that AN446 is superior to VPA in its anticancer activities, without exhibiting higher toxicity. Its ability to specifically target cancer cells can be attributed to its properties as an ester prodrug that is activated by cellular esterases to release the active drug(s) [45,46]. The higher level of cellular esterases found in cancer cells compared to normal cells offers a promising strategy to minimize the toxic effects of the AN446 metabolites on normal tissues [47]. Altogether, the combination treatment of AN446 and doxorubicin (AN446 + Dox) differentially affect cancer cells compared to normal cells. Inhibition of HDAC activity in cancer cells by AN446 combined with Dox leads to un-repressed chromatin and change in gene expression along with DNA breaks and reduced DNA repair. The high levels of collagen I, C-MYC, SIRT1, COX2, and RAD51 expression in the tumor are attenuated by the treatment, leading to reduced fibrosis, viability, inflammation, culminating in cancer cell death. In normal cells, in particular those residing in the heart, DOX-induced toxicity that results in fibrosis, DNA breaks, and inflammation is reduced by the addition of AN446 that increases DNA repair, SOD1, SIRT1, and c-MYC expression, altogether prolonging cell survival. Therefore, AN446 proved to be a promising therapeutic answer to the unmet needs of TNBC treatment.

## Figures and Tables

**Figure 1 pharmaceuticals-14-01244-f001:**
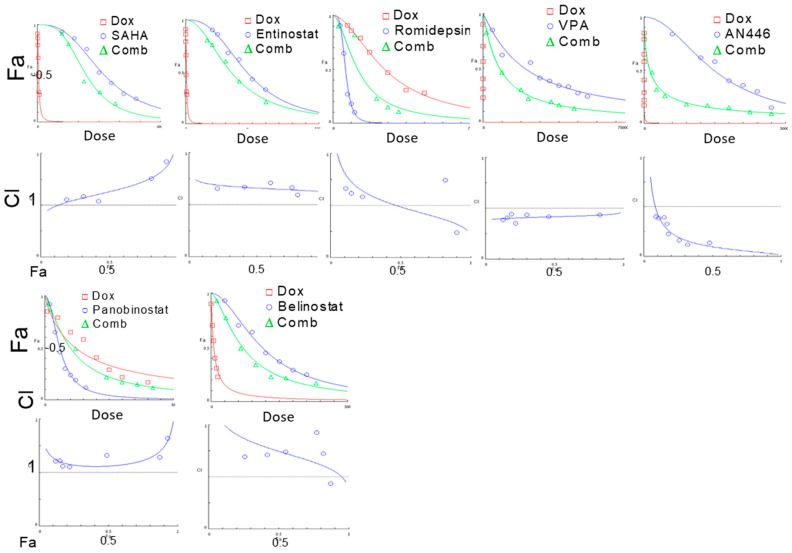
Effect of HDACIs, Dox, or their combinations on the viability of 4T1 murine breast cell line. 4T1 cells (4 × 10^3^/well) were seeded in 96 well-plates, incubated overnight, and then treated with SAHA (0.1–3 µM), entinostat (0.1–3 μM), Rom (0.1–3 nM), VPA (1–10 mM), AN446 (10–60 μM), panobinostat (1–20 nM), belinostat (50–400 nM), or Dox (1–40 nM), or their IC_50_ combination. After 72 h of treatment. the viability of the cells was determined by the Hoechst assay. The average values of three independent experiments are presented. The drug concentrations-dependence plots were generated for each of the drugs alone and in combination with Dox. The fraction of affected cells (Fa) as a function of Dox concentrations or HDAC inhibitor concentrations as a single agent and their combination are presented; CIs as a function of Fa in 4T1 cells were produced by CompuSyn software are shown. The IC_50_ value of each drug as a single agent and in combination with Dox and the calculated combination index (CI) are shown. Each treatment was performed in triplicate in three independent experiments and is presented as the mean ± SE.

**Figure 2 pharmaceuticals-14-01244-f002:**
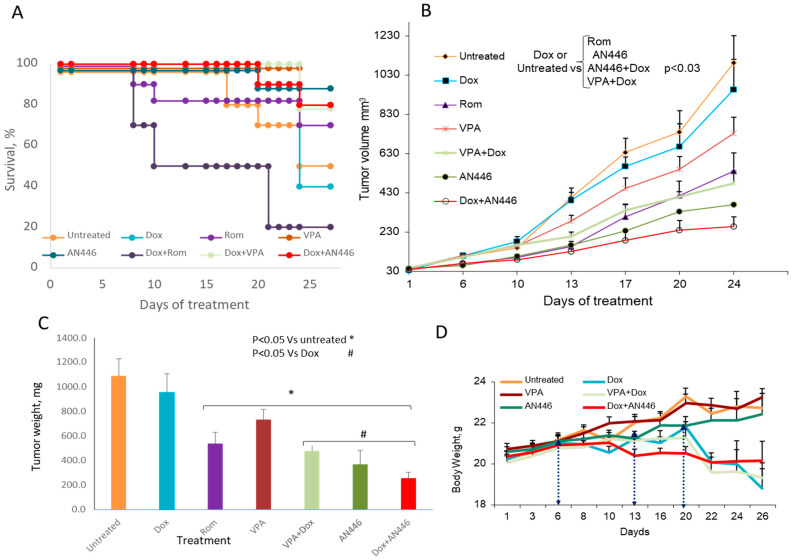
Effect of HDACIs, Dox, and their combinations in the 4T1 murine mammary carcinoma model. Eight-week-old BALB/c mice were inoculated sc with 5 × 10^4^ 4T1 breast cancer cells. When tumor volume reached 30–40 mm^3^, the mice were assigned blindly to the following treatments: saline ip (*n* = 5) or po (*n* = 5); 4 mg/kg ip Dox, once/week (*n* = 10); 25 mg/kg po AN446 thrice/week (*n* = 10); 2 mg/kg ip Rom the first week and 1 mg/kg the second and third weeks (*n* = 10); 200 mg/kg po VPA thrice/week (*n* = 10); 25 mg/kg po AN446 + 4 mg/kg ip Dox (*n* = 10); 2 mg/kg ip Rom the first week and 1 mg/kg the second and third weeks. Rom doses were given with +4 mg/kg ip Dox (*n* = 10); 200 mg/kg po VPA + 4 mg/kg ip Dox (*n* = 10). The percent of surviving mice was assessed by a Kaplan–Meier graph (**A**). Tumor growth as a function of time (mean ± SE, mm^3^) was plotted and its growth among groups was compared by one-way ANOVA (**B**). The average tumor weight at the termination point is shown (**C**). Changes in average tumor weight along time lines, the time-points (**D**). * Significantly changed from vehicle control; # significantly changed from Dox treatment.

**Figure 3 pharmaceuticals-14-01244-f003:**
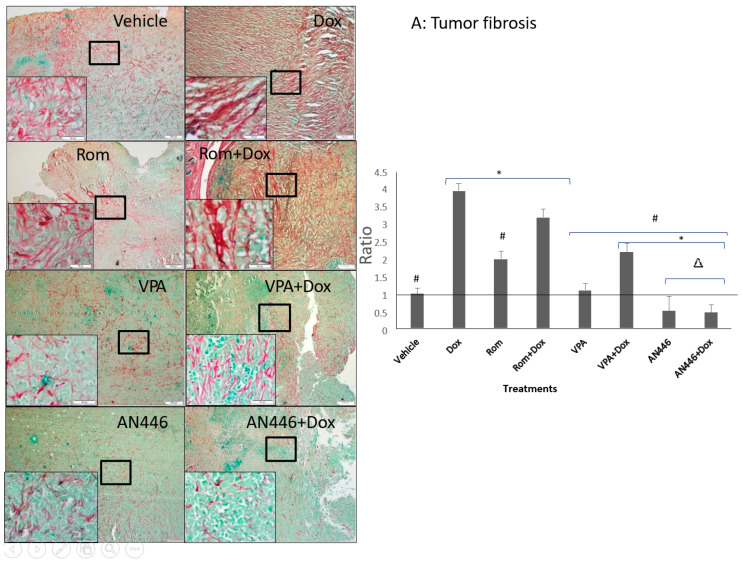
Effect of HDACIs, Dox and their combinations on fibrosis in tumors (**A**), hearts (**B**), and kidneys (**C**). Sections of organs were stained with Picrosirius red and fast green for fibrous collagen for the visualization of interstitial fibrosis. Bar = 200 μm, the 4-fold magnified picture with 10-fold magnified in the inserts, taken from the area indicated by the square. Quantifying fibrosis was performed on four different fields/sections taken from three mice/groups. tumors (**A**), hearts, CM = cardiomyocytes; CF = cardio-fibrosis (**B**), kidneys, NF = nephron-fibrosis (**C**). Statistical analyses were performed with graph prism 7.3. Outliners were identified using * *p* < 0.05 for all drugs treated vs. vehicle-treated mice; # *p* < 0.05 drug-treated vs. Dox-treated; ∆ *p* < 0.05, drug-treated vs. Rom + Dox-treated mice.

**Figure 4 pharmaceuticals-14-01244-f004:**
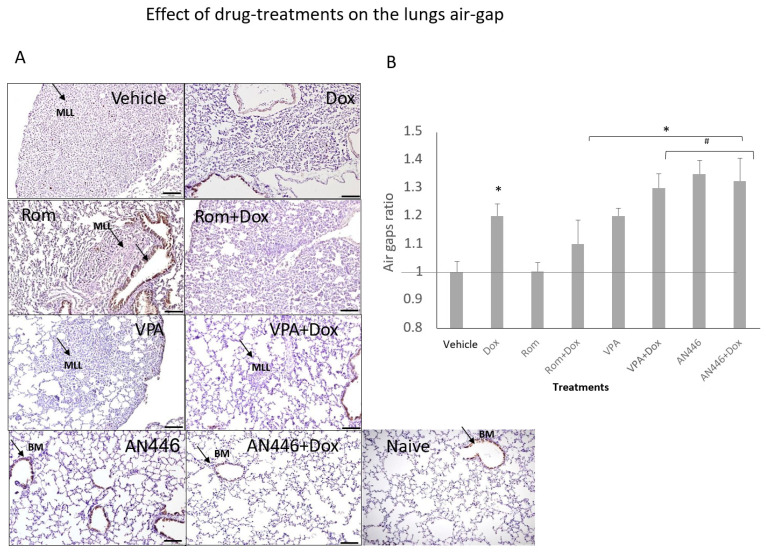
Effect of HDACIs, Dox, and their combinations on air space in the lungs. Sections of lungs (**A**) were stained (DAB) for Ki-67 and counterstained with H&E. Bar = 200 μm (**B**). BM = basal membrane; MLL = metastatic lung lesions. Image analysis was conducted by ImageJ. The air gaps were evaluated as described above and plotted as a ratio of treatment to vehicle (**B**). Adjusted *p* was applied with Dunn’s test for multiple compression correction. The results represent the average ± SEM of 12 different fields, * *p* < 0.05 all drugs treated vs. vehicle-treated mice; # *p* < 0.05 drug-treated vs. Dox-treated. Lung fibrosis, Sections of the lungs were stained with Picrosirius red and fast green (**C**).

**Figure 5 pharmaceuticals-14-01244-f005:**
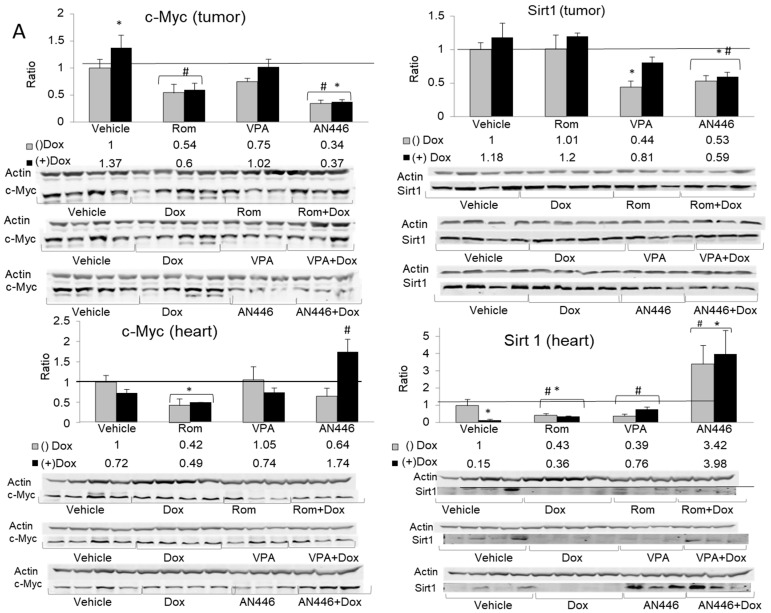
Effect of treatment with HDACIs, Dox, and their combination on protein expression in the tumors and the hearts of mice bearing 4T1 breast cancer. For the detection of the specified proteins, samples were resolved on SDS gels by Western blot analysis. Lysates of tumors (*n* = 3–4) (20 µg protein) or hearts (*n* = 3–4) (30 µg protein) were loaded on the gels and subjected to the analyses using the specific antibodies for the detection c-Myc (**left**-hand side) and SIRT 1 (**right**-hand side), resolved on 10% SDS gel (**A**); detection of SOD1 (**left**-hand side) and COX2 (**right**-hand side) resolved on 12% SDS gel (**B**); fold increase represents the ratio of band intensity (mean ± SE) of drug-treated to vehicle-treated, each normalized to actin signal, * *p* < 0.05 all drug-treated vs. vehicle-treated mice; # *p* < 0.05 Dox-treated vs. drug-treated mice.

**Figure 6 pharmaceuticals-14-01244-f006:**
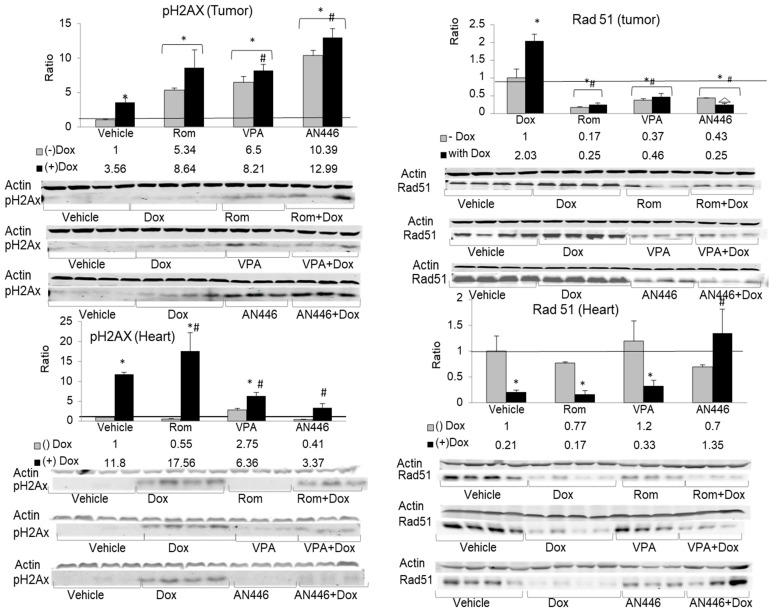
Effect of the treatments on DNA damage and repair response. Lysate samples for detection of DNA DSBs were stained for γH2AX (**left**-hand side) and the markers for the detection of RAD51 were stained for anti-RAD51 (**right**-hand side); The peptides were resolved on 15% SDS gel. Fold increase was determined as described in Figure 5 above. * *p* < 0.05 all drug-treated vs. vehicle-treated mice; # *p* < 0.05 Dox-treated vs. drug-treated mice.

**Table 1 pharmaceuticals-14-01244-t001:** Values of IC_50_ for each of the indicated drug as a single agent and in combination with Dox.

Drugs	IC_50_, Single Agent	Drugs Ratio (Dox:Drug)	HDAC IC_50_ in Combination	CI
Dox, nM	18.2 ± 3.7			
SAHA, µM	1.1 ± 0.2	1:75	0.7 ± 0.2	1.3
Entinostat, µM	1.8 ± 0.2	1:100	1.3 ± 0.2	1.6
Romidepsin, nM	6.0 ± 0.8	2:1	2.9 ± 0.7	1
VPA, mM	2.8 ± 0.4	1:200,000	1.3 ± 0.2	0.9
Panobinostat nM	6.7 ± 1.6	2:1	3.1 ± 0.4	1.1
Belinostat, nM	217 ± 34	1:10	158 ± 38	1.6
AN446	29 ± 1.9	29.3:1.9	3.0 ± 0.7	0.6

The ratio of the drugs used in the combination was their IC_50_ ratio. The Cis was determined by the application and represents the concentration ratio of the drugs as single agents and in the combination (com) needed to achieve 50% survival.

## Data Availability

Data is contained within the article.

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
