# Peer review of "Valproic Acid Prodrug Affects Selective Markers, Augments Doxorubicin Anticancer Activity and Attenuates Its Toxicity in a Murine Model of Aggressive Breast Cancer"

_pharmaceuticals, 2021, doi:10.3390/ph14121244_

Round 1

Reviewer 1 Report

Major:

  1. Why did you use different drug ratios depending on the drug used? How were ratios chosen?
  2. It is recommended to perform an isobolographic analysis to assess types of interactions between drugs using a wide range of concentrations, not just randomly selected doses.
  3. How the therapeutic doses of drugs used in the mouse model were calculated?

Minor:

  1. English language correction is needed.
  2. „Ip.” and „io.” abbreviations should be explained when the first time used.
  3. Line 273 - punctuation errors.

Author Response

Major:

  1. Why did you use different drug ratios depending on the drug used? How were ratios chosen?
  2. It is recommended to perform an isobolographic analysis to assess types of interactions between drugs using a wide range of concentrations, not just randomly selected doses.

1 & 2. Isobolographic analysis is a good and valid method to determine drug interaction. We choose a different valid and acceptable method for analyzing drug interaction named Median Effect Analysis (MEA) as specified under 2.7 and referred to the publication (reference 22) and the software available online.  

.

  1. How the therapeutic doses of drugs used in the mouse model were calculated?

Under 3.2 section we wrote: " Based on our previous studies [8-11], vast experience, literature reports and preliminary experiments to examine efficacy and toxicity of the tested drugs, the efficacious and maximum tolerated drugs doses were chosen for this main experiments."

Minor:

  1. English language correction is needed.
  2. „Ip.” and „io.” abbreviations should be explained when the first time used.
  3. Line 273 - punctuation errors.

The minor comments were corrected.

Reviewer 2 Report

The manuscript by Tarasenkoa et al, studies the synergistic anti-cancer effect of Valproic acid prodrug with doxorubicin. The combinatorial anti-cancer treatment regimens with histone deacetylase inhibitors (HDACIs) is an important area of research in oncology and specifically in triple negative breast cancers. While manuscript is reasonably well-written there are some areas of concern which need to be addressed.

1) Atleast in the version provided to me,  Figure legends and details were missing for all figures. Some of the figures do not even have figure captions.

2) The combination index (CI) cut-offs for additive, synergistic and anatagonistic mechanisms need to be explicitly mentions in the methods section.

3) For Figure 2C only 2 tumors were included in statistical analysis for (Rom+Dox). This would make me wonder if the statistical analysis was even possible?! For figure 2B, what kind of statistical analysis was performed for tumor growth kinetics? Was it multi-time point kinetic analysis or end-point analysis? Either way, provide more details on the statistical analysis (simply stating 2-way ANOVA) is not sufficient, what type of adjustments were made for decreased survival from ne-time point to other time-point. It seems from Fig2A atleast 4 cohorts had moe than 50% mortatlity by the end –point. Further (although minor comment), the color coding for fig 2 A and 2B should be uniform or readers to easily compare.

4) As figure 3 had no legends it was difficult to follow the authors’ narrative. While authors’ seem to indicate cardiac tissue for figure 3B, the magnification of the image does not provide a clear outline of myocardium. Higher magnification images showing the delineation between myocardium and fibrous tissue need to be provided. Similarly for figure 3C, the glomerular structure of kidney and fibrous deposition is unclear. Higher magnification images need to be provided. Along the lines, for figure 4, it is unclear if the lung fibrosis is along the lung parenchyma or the bronchioles. Figure 4C seem to indicate inflammatory injury more than fibrosis.

5) In figure 5 and 6, the authors have performed expression analysis of SOD, COX2 and DNA damage/repair enzymes. However, what is the synergistic mechanism of valproate with doxorubicin? Atleast in figure 5A, it seems there is no synergistic inhibition of cMyc and SIRT1 expression with any combination regimens.

6) Doxorbicin is a cell cycle inhibitor. Does combination of HDACIs modifiy the cell cycle. If so, which phase of cell cycle.

7) As fibrosis is reduced, it seems like the combination is inhibiting cancer associated fibroblasts (CAFs). What is the drug cytotoxicity index on CAFs? This is a simple cell culture study but is very important as the authors’ swem to emphasize that their drug combination is reducing fibrosis.

Author Response

The manuscript by Tarasenkoa et al, studies the synergistic anti-cancer effect of Valproic acid prodrug with doxorubicin. The combinatorial anti-cancer treatment regimens with histone deacetylase inhibitors (HDACIs) is an important area of research in oncology and specifically in triple negative breast cancers. While manuscript is reasonably well-written there are some areas of concern which need to be addressed.

  • Atleast in the version provided to me, Figure legends and details were missing for all figures. Some of the figures do not even have figure captions.

All the captions and Legends are in place.

  • The combination index (CI) cut-offs for additive, synergistic and anatagonistic mechanisms need to be explicitly mentions in the methods section.

See also the above response to Reviewer 1. Additional explanations were added to 2.7 section, under data analysis. Reference 22 explains the method in detail.

  • For Figure 2C only 2 tumors were included in statistical analysis for (Rom+Dox). This would make me wonder if the statistical analysis was even possible

Rom+Dox was omitted from the graphs due to treatment toxicity where the animals in this group reached the endpoint early in the study.

  • For figure 2B, what kind of statistical analysis was performed for tumor growth kinetics? Was it multi-time point kinetic analysis or end-point analysis? Either way, provide more details on the statistical analysis (simply stating 2-way ANOVA) is not sufficient, what type of adjustments were made for decreased survival from ne-time point to other time-point.

It was done with two-way ANOVA for repeated measures with multi-time point kinetic analysis done using Prism software.

It seems from Fig2A atleast 4 cohorts had moe than 50% mortatlity by the end –point. Three cohorts had 50% mortality by the endpoint, vehicle-, Rom-, and Rom+Dox- -treated.

5) Further (although minor comment), the color coding for fig 2 A and 2B should be uniform or readers to easily compare. The colors were adjusted

As figure 3 had no legends it was difficult to follow the authors’ narrative. While authors’ seem to indicate cardiac tissue for figure 3B, the mendagnification of the image does not provide a clear outline of myocardium. Higher magnification images showing the delineation between myocardium and fibrous tissue need to be provided. Similarly for figure 3C, the glomerular structure of kidney and fibrous deposition is unclear. Higher magnification images need to be provided.

The validity of the morphology was confirmed by experts. The magnification shown in the figure is representative of those used to perform the image analysis. A large area and multiple pictures were required for the analysis. All the legends were provided before the references. We added arrows to the pictures to ease the identification of the tissue elements.

Along the lines, for figure 4, it is unclear if the lung fibrosis is along the lung parenchyma or the bronchioles. Figure 4C seem to indicate inflammatory injury more than fibrosis. In this figure we see lung metastases along with deposit of collagen, most likely contributed by cancer associated fibroblasts.

  • In figure 5 and 6, the authors have performed expression analysis of SOD, COX2 and DNA damage/repair enzymes. However, what is the synergistic mechanism of valproate with doxorubicin? Atleast in figure 5A, it seems there is no synergistic inhibition of cMyc and SIRT1 expression with any combination regimens.

When we observe a synergistic effect on the cells, it is not the outcome of a change in a single gene, it is rather the integration of all the changes in multiple genes that produce the synergy.

As we have shown, dramatic changes of genes that modulate; inflammation, DNA damage and response, etc. occurred following the combination treatment and they all contribute to the observed synergistic effect. 

  • Doxorbicin is a cell cycle inhibitor. Does combination of HDACIs modifiy the cell cycle. If so, which phase of cell cycle.

The cell cycle is not the focus of our investigations. Most studies indicate that HDAC inhibitors arrest the cell cycle at G1. Dox arrest cell cycle at G2M, indicating that they affect different targets in the cell cycle and therefore this difference can also contribute to their synergistic interaction. 

  • As fibrosis is reduced, it seems like the combination is inhibiting cancer associated fibroblasts (CAFs). What is the drug cytotoxicity index on CAFs? This is a simple cell culture study but is very important as the authors’ swem to emphasize that their drug combination is reducing.

This is an important point, it was not in the scope of this work and it is not a simple experiment. The tumor needs to be separated in tissue culture to its different cell types including cancer-associated fibroblasts and tumor cells and then titrate the cytotoxic effect of the drug combination. We have performed cardiomyoblasts and cardiofibroblast separation previously and shown that they are differentially affected by HDAC inhibitors (British journal of cancer 96 (11), 1667-1674. 2007)

Reviewer 3 Report

In this research manuscript (Pharmaceuticlas-1427404) titled “Valproic acid prodrug affects selective marker, augments doxorubicin anticancer activity and attenuates its toxicity in murine model of aggressive breast cancer” by Tarasenko, H. et al., the authors demonstrated synergistic anticancer effects of AN446 + Doxorubicin in 4T1 syngenetic graft model. The authors showed 1) Synergism of AN446 with Doxorubicin in murine TNBC cells 2) Kaplan-Mier plot with tumor size measurement 3) Analysis of fibrosis in tumors, hearts and kidneys 4) Measurement of lung metastasis along with lung fibrosis 5) Biochemical changes of several key markers including c-MYC, SIRT1, SOD2, COX2, pH2AX and RAD51 in both tumor and heart tissues.

To make their manuscript more suitable for publication, the authors need to address the following points.

Major points

  1. (Self-plagiarism) The “2.4 Immunohisochemistry”, “2.6 Western blot analysis section” are reused from previous author’s publication. It is necessary to get opinion from the Pharmaceuticals’ editor whether the reusing of this part has no ethical violation.
  2. The author performed an in vivo experiment using only one TNBC cell line, some of the results of this Ms. overlaps with those of the author's previous research papers published in the past (Invest New Drugs (2017), Biochem Pharmacol (2014)). The effects of AN466+Dox combination in 4T1 cells should be verified in human TNBC cells at least in vitro. These results will support the results of this manuscript.
  3. In the animal study, it would be helpful to include normal control to compare the fibrosis and air gaps.
  4. In the discussion, there is not much content except for the re-description of result. Please add more for the differential regulation of MYC, SIRT1, SOD, COX2, and RAD51, H2AX by AN466+Dox in tumor and normal tissue. What is the mechanism of these differential regulation? Are there any previous studies?

Minor points

  1. Titles & Legends of Figure 6 and Table 1 are missing.

The authors presented the legends of Figure 5A – 5E, I can’t find Figure 5C – E in manuscript.

Please rewrite the Figure 5 legends. Does Figure 6 exist?

  1. Line 39, 194, 216, 262, change “in vitro” and “in vivo” to “in vitro” and “in vivo”.
  2. Lines 39 – 41, Currently four HDACis have been approved by the US FDA (Autin, P., Blanquart, C. & Fradin, D. Epigenetic Drugs for Cancer and microRNAs: A Focus on Histone Deacetylase Inhibitors. Cancers 11, 1530 (2019)). Please update this.
  3. Lines 93 – 94, Doxorubicin hydrochloride from Ebewe Pharma is dissolved in NaCl-containing water (https://www.medsafe.govt.nz/Consumers/cmi/d/doxorubicinEbewe.pdf). Did you add proper control solution in your experiments? Please justify this.
  4. Lines 95 – 100, please add catalog numbers of all antibodies used in this study.
  5. Many typos in the expression of superscript. Please update this.
  6. Please check author’s name carefully.
  7. Line 218, What is the “sc” 4T1 cells?
  8. Lines 219 – 220. It would be better to show the graph for the change of body weights of all groups.
  9. Lines 222 – 224, Is this statistically insignificant? AN446 alone 90% survived vs. AN446+Dox 80% survived.
  10. Line 274, ,”which” ?
  11. Fig. 5A. needs statistical analysis.
  12. Lines 391 – 393, What is the background for the relationship of COX2 downregulation with Dox-induced fibrosis?
  13. Line 475, MTE à EMT
  14. Line 639, HO1 should be SOD1.
  15. Overall, the quality of Figures are poor, at least the brightness and shape of the picture to be unified.
  16. Authors need to put some marks (arrows, arrow heads) into (immuno)-histochemical staining photos with additional explanation.
  17. Authors need to put additional comments on the increased expression of c-Myc in the heart by Dox+AN446 in discussion.

Author Response

In this research manuscript (Pharmaceuticlas-1427404) titled “Valproic acid prodrug affects selective marker, augments doxorubicin anticancer activity and attenuates its toxicity in murine model of aggressive breast cancer” by Tarasenko, H. et al., the authors demonstrated synergistic anticancer effects of AN446 + Doxorubicin in 4T1 syngenetic graft model. The authors showed 1) Synergism of AN446 with Doxorubicin in murine TNBC cells 2) Kaplan-Mier plot with tumor size measurement 3) Analysis of fibrosis in tumors, hearts and kidneys 4) Measurement of lung metastasis along with lung fibrosis 5) Biochemical changes of several key markers including c-MYC, SIRT1, SOD2, COX2, pH2AX and RAD51 in both tumor and heart tissues.

To make their manuscript more suitable for publication, the authors need to address the following points.

Major points

  1. (Self-plagiarism) The “2.4 Immunohisochemistry”, “2.6 Western blot analysis section” are reused from previous author’s publication. It is necessary to get opinion from the Pharmaceuticals’ editor whether the reusing of this part has no ethical violation.

"Self-plagiarism" in a way is a contradiction in terms. A great deal of time passed between the two publications and the instruments, the antibodies and some details of the methods were changed.

  1. The author performed an in vivo experiment using only one TNBC cell line, some of the results of this Ms. overlaps with those of the author's previous research papers published in the past (Invest New Drugs (2017), Biochem Pharmacol (2014)).

 No overlapping data were presented in the indicated papers. In Biochem Pharmacol (2014), the in vivo model used was glioblastoma AN446 that was compared to AN7 for mostly expression of different protein markers. In Invest New Drugs (2017) we described different studies where the emphasis of the study was on changes of specific methylated and acetylated histone in the glioblastoma U251 cells, in human astrocytes and in U251 tumors following drug treatments.

The effects of AN466+Dox combination in 4T1 cells should be verified in human TNBC cells at least in vitro. These results will support the results of this manuscript. In our paper [ref. 12 in the present manuscript]. Comparison of the anticancer properties of a novel valproic acid prodrug to leading histone deacetylase inhibitors, 2017, we have performed a detailed analysis comparing the interaction between Dox and numbers of HDACIs glioblastoma cell line and the human TNBC, MD-MA-231. 

  1. In the animal study, it would be helpful to include normal control to compare the fibrosis and air gaps. Was added.
  2. In the discussion, there is not much content except for the re-description of result. Please add more for the differential regulation of, MYC, SIRT1, SOD, COX2, and RAD51, H2AX by AN466+Dox in tumor and normal tissue. What is the mechanism of these differential regulation? Are there any previous studies?    

    We added more discussion mainly to MYC function. The possible functions of the genes are discussed here in three different paragraphs before the last one in the discussion.

Minor points

  1. Titles & Legends of Figure 6 and Table 1 are missing. Provided

The authors presented the legends of Figure 5A – 5E, I can’t find Figure 5C – E in manuscript. Please rewrite the Figure 5 legends. Does Figure 6 exist?

All is in order and there is Fig. 6

  • Line 39, 194, 216, 262, change “in vitro” and “in vivo” to “in vitro” and “in vivo”. Done

Lines 39 – 41, Currently four HDACis have been approved by the US FDA (Autin, P., Blanquart, C. & Fradin, D. Epigenetic Drugs for Cancer and microRNAs: A Focus on Histone Deacetylase Inhibitors. Cancers 11, 1530 (2019)). Please update this.   The reference was added and the line was inserted.

  1. Lines 93 – 94, Doxorubicin hydrochloride from Ebewe Pharma is dissolved in NaCl-containing water (https://www.medsafe.govt.nz/Consumers/cmi/d/doxorubicinEbewe.pdf). Did you add proper control solution in your experiments? Please justify this. Doxorubicin hydrochloride from Ebewe Pharma was diluted in saline solution (0.9% NaCl). This was added under "Materials and Methods". In the vehicle control group, half of the animal received gavage and the other half of them received the vehicle ip (as described under the legend to Fig. 2A.
  2. Lines 95 – 100, please add catalog numbers of all antibodies used in this study. Catalog numbers of all antibodies used in this study. Were added.
  3. Many typos in the expression of superscript. Please update this. Corrected
  4. Please check author’s name carefully. Done
  5. Line 218, What is the “sc” 4T1 cells? Balb-c mice implanted sc (subcutaneously) with 4T1 cells. Done
  6. Lines 219 – 220. It would be better to show the graph for the change of body weights of all groups. The graph was produced
  1. Lines 222 – 224, Is this statistically insignificant? AN446 alone 90% survived vs. AN446+Dox 80% survived. Certainly not! That is the point, the group of mice treated with AN446 alone, 90% survived and in AN446+Dox treated group 80% survived. However, treatment with Dox alone resulted in 40% survival. The comparison is between Dox toxicity and the protective effect of AN446 when added to Dox treatment was demonstrated.
  2. Line 274, ,”which” ? The question is unclear
  1. Fig. 5A. needs statistical analysis. By-nature Western-Blot analysis is semi-quantitative. We have run multiple samples from the hearts and the tumors of the mice their fold increased expression was calculated as described in lines 146-160. T-test was performed comparing the fold expression of treated mice to that of control mice, significant difference to control was marked by (*), and a significant difference to Dox treatment was marked (#). All these are shown in the figure and explained in the legends to the figures.
  2. Lines 391 – 393, What is the background for the relationship of COX2 downregulation with Dox-induced fibrosis? A correlation is observed between reduction in Cox2 expression and reduction in fibrosis, by AN446 or AN446+Dox in the hearts (compared to that in Dox-treated mice) and in the tumors (compared to that in untreated mice or Dox treated mice). It was suggested that COX-2 expression may play an important role as a biomarker for estimating the aggressiveness of breast cancer in clinical practice [ref 29].
  3. Line 475, MTE à EMT. Done
  4. Line 639, HO1 should be SOD1. Corrected.
  5. Overall, the quality of Figures are poor, at least the brightness and shape of the picture to be unified. We believe the final production would look better.
  6. Authors need to put some marks (arrows, arrow heads) into (immuno)-histochemical staining photos with additional explanation. Done

18    Authors need to put additional comments on the increased expression of c-Myc in the heart by Dox+AN446 in discussion. We added it in lines 693-8 in the marked manuscript: " Previously we have shown that HDAC inhibitor elevated angiogenesis in the heart by downregulating the expression of the c-Myc-regulated angiogenesis inhibitor TSP-1 and co-elevated the expression of the proangiogenic factors FGF-2 and VEGF [8]. Similarly, we have shown that addition of AN446 to Dox treatment, elevated angiogenesis in the heart by effectively and by a tissue specific-manner increased FGF-2 and VEGF expression [10], further supporting its cardioprotective function shown in this study."

Round 2

Reviewer 1 Report

The authors responded to the comments. 

Author Response

The authors responded to the comments. 

Reviewer 3 Report

To avoid plariarism and copyright conflict with other journals arising from the re-use of previous author's article. please add ref. of prev author's article containing same sentences in Line 144.

To avoid plariarism and copyright conflict with other journals arsing from the re-use of previous author's article. please add ref. of prev author's article containing same sentences in Line 165.

Line 391 Author needs to put citations of previous research articles, in which author already carried out comparing the expression of c-Myc/p-c-Myc between AN446 and AN446 + Dox in heart tissue (ref 10).

Line 529 Author needs to put citations of previous research articles, in which author already compared the difference of expression of c-Myc/p-c-Myc between AN446 and AN446 + Dox in heart tissue (ref 10).

Regarding synergistic effect between AN446 and Dox, author need to put citations of previous research articles (ref 10), in which author already tested synergistic effect in 4T1 cells.

Regarding the results presented in  the lower panel of Fig6. authors already observed the difference of expression levels of Rad51 and pH2AX in mice (ref 12), author need to put citations and comments on it.

AN446 contains two possible valproate groups and Acyclovir linked by amide and ester bonds, how about to put additional comments on the assuming mechanism of metabolized product of AN446 which shows relatively high synergistic effect in cancer cells and protective effect in heart in discussion?

Did you compare the differences of effectiveness among VPA, VPA-Acylovir ester, Valpromide-Acyclorvir and AN446 in any conditions?

In the discussion, please add more for the differential regulation of MYC (done?), SIRT1, SOD, COX2, and RAD51, H2AX by AN466+Dox in tumor and normal tissue.

What is the mechanism of these differential regulation? Are there any previous studies?

Author Response

√To avoid plariarism and copyright conflict with other journals arising from the re-use of previous author's article. please add ref. of prev author's article containing same sentences in Line 144. Done

√To avoid plariarism and copyright conflict with other journals arsing from the re-use of previous author's article. please add ref. of prev author's article containing same sentences in Line 165. Done

√Line 391 Author needs to put citations of previous research articles, in which author already carried out comparing the expression of c-Myc/p-c-Myc between AN446 and AN446 + Dox in heart tissue (ref 10). Done

√Line 529 Author needs to put citations of previous research articles, in which author already compared the difference of expression of c-Myc/p-c-Myc between AN446 and AN446 + Dox in heart tissue (ref 10). Done

√Regarding synergistic effect between AN446 and Dox, author need to put citations of previous research articles (ref 10), in which author already tested synergistic effect in 4T1 cells. Done

Regarding the results presented in the lower panel of Fig6. authors already observed the difference of expression levels of Rad51 and pH2AX in mice (ref 12), author need to put citations and comments on it. high synergistic effect in cancer cells and protective effect in heart in discussion? Answer: In ref 12 we examine the effect of AN446 on cell lines: U251 and the cardiomyoblasts H9c2 cell lines.

Did you compare the differences of effectiveness among VPA, VPA-Acylovir ester, Valpromide-Acyclorvir and AN446 in any conditions?

Answer: This is described in reference 9.

Thank you for the question. In reference 9 we have described the structure-activity-study of VPA derivatives. We added the following paragraph prior to reference 9: “Next, we synthesized and tested VPA derivatives as anticancer agents. AN452 the VPA ester of acyclovir, AN463 the VPA amide (valpromide) of acyclovir and AN446 the amide-ester (valpromide-ester) of acyclovir. AN463, shown to be inactive and the ester AN452 was 5-10 fold less active than AN446 [9].”  

In the discussion, please add more for the differential regulation of MYC (done?), SIRT1, SOD, COX2, and RAD51, H2AX by AN466+Dox in tumor and normal tissue.

What is the mechanism of these differential regulation? Are there any previous studies?

The last two comments are subject to future studies.